# Periodic-Filtering Method for Low-SNR Vibration Radar Signal

Yun Lin [ID], Linghan Zhang, Hongwei Han, Yang Li, Wenjie Shen [ID] and Yanping Wang *

Radar Monitoring Technology Laboratory, School of Information Science and Technology, North China University of Technology, Beijing 100144, China
* Correspondence: wangyp@ncut.edu.cn

**Abstract:** Radar is a non-contact, high-precision vibration measurement device and an important tool for bridge vibration monitoring. Vibration information needs to be extracted from the radar phase, but the radar phase information is sensitive to noise. Under low signal-to-noise ratio (SNR) data acquisition conditions, such as low radar transmission power or a long observation distance, differential phase jump errors occur and clutter estimation becomes difficult, which leads to inaccurate inversion of vibration deformation. Traditional low-pass filtering methods can filter out noise to improve SNR, but they require oversampling. The sampling rate needs to be several times higher than the Doppler bandwidth, which is several times higher than the vibration frequency. This puts high data acquisition requirements on radar systems and causes large data volumes. Therefore, this paper proposes a novel vibration signal filtering method called the periodic filtering method. The method uses the periodicity feature of vibration signals for filtering without oversampling. This paper derives the time-domain and frequency-domain expressions for the periodic filter and presents a deformation inversion process based on them. The process involves extracting the vibration frequency in the Doppler domain, suppressing noise through periodic filtering, estimating clutter using circle fitting on the data complex plane, and inverting final deformation with differential phase. The method is verified through simulation experiments, calibration experiments, and bridge vibration experiments. The results show that the new periodic filtering method can improve the SNR by five times, resolve differential phase jumps, and accurately estimate clutter, thus getting submillimeter-level vibration deformation at low SNR.

**Keywords:** radar; vibration; filtering; clutter suppression

## 1. Introduction

Vibration is a key indicator of the health of infrastructure such as bridges and buildings [1–4]. There are many methods to measure vibration, which fall into two main categories: contact and non-contact. Contact methods use sensors such as displacement, velocity, and acceleration sensors that need to be attached to the vibrating point. However, these methods may not be feasible in some situations due to the constraints of the installation location. Non-contact methods, such as lidar and radar, do not require installation. They can be placed at a suitable distance and adjust their beams to target the vibrating points. Lidar has high precision, but it is susceptible to environmental factors such as rain and fog. Radar is immune to these factors and can operate in any weather condition. Radar works by transmitting microwave signals and receiving the target echo. It analyzes the phase change of the echo signal over time to obtain the vibration deformation, which has very high submillimeter-level measurement accuracy [5,6]. Therefore, radar has become more popular in the field of infrastructure vibration monitoring due to its advantages of being all-day, all-weather, high-precision, and non-contact.

Radar-based vibration measurement has been a research topic for over 20 years. Tarchi et al. first proposed using radar interferometry to monitor buildings in 2000 [7]. Piraccini et al. developed and tested the first radar system for bridge dynamic monitoring in 2004 [8]. The University of Florence and IDS (Ingegneria dei Sistemi SpA) collaborated to develop

the IBIS series of systems in 2007 [9], one of which, the IBIS-S system, became widely used for infrastructure vibration monitoring [1,3,5,10,11]. This system operates in the Ku band with a power of 40 W. Vibration radar systems have improved in recent years. In 2020, the Beijing Institute of Technology created a multi-channel radar that can be used for imaging and vibration information acquisition of large infrastructures [12–14]. Since 2018, small and lightweight millimeter-wave radars have been used for vibration monitoring. The Chinese Academy of Sciences made a Ka-band radar system for measuring bridge vibration [15]. Chengkun Jiang et al. from Tsinghua University measured the vibration of industrial systems using a commercial millimeter-wave board, the Texas Instruments (TI) IWR1642 BoosterPack [16]. The IWR1642 operates at 77 GHZ frequency with a power of only 17.8 mv. The authors of [17,18] used a TI millimeter-wave MIMO board to monitor bridges and vehicles. Low-cost millimeter-wave radars may offer new possibilities for infrastructure vibration monitoring.

Radar-based vibration measurement is challenging. The echo's differential phase contains the deformation information, but it is easily affected by noise and clutter [2,12,19–21]. The radar transmission power must meet certain requirements, and the observation distance is usually short. When the vibrating point has weak scattering, the strong corner reflector at the measured point can improve the signal-to-noise ratio (SNR) and signal-to-clutter ratio (SCR). The authors of [1,22] used this method of installing corner reflectors. The authors of [11] did not install a corner reflector when measuring the bridge bottom deformation, because the support structure under the bridge is a good corner reflector with strong scattering. This helps extract deformation information accurately. However, installing corner reflectors makes the measurement task more difficult and inconvenient. When the observation distance is long, the radar power is low, or the vibrating target has weak scattering, the echo signal has a low SNR. This causes large differential phase noise and even jump errors [23]. Under low SNR conditions, the phase wrapping issue randomly generates jump errors in differential phase that cannot be unwrapped using the phase continuity feature. Moreover, environmental clutter adds to the signal, so that the differential phase does not accurately reflect deformation information.

To address these challenges, researchers have used signal processing to improve the SNR and SCR of signals. For clutter suppression, the most classic method is the circle fitting method, which [12,16,21,22,24] used. In the complex plane, the radar signal forms an arc or a circle, and the clutter causes the circle center to deviate from the origin. By estimating the circle center, the signal origin can return to the circle center to eliminate the clutter effect. For noise suppression, the classic algorithm is a low-pass filtering method. Different papers use low-pass filters at different stages of the processing process. The authors of [21] applied low-pass filtering to the original echo data at the beginning, while [6,22] applied low-pass filtering to the final deformation to remove high-frequency noise. The former can solve the phase jump problem under low SNR and estimate the clutter more accurately, but it requires oversampling. The sampling rate needs to be several times the Doppler bandwidth, and the harmonic characteristics of the Doppler signal make its bandwidth several times the vibration frequency [9], which demands higher data acquisition and produces more data. However, the latter does not require oversampling because the bandwidth of the deformation variable is smaller than that of the original signal, but it cannot solve the phase jump problem and is not conducive to clutter estimation. The author used the TI radar board for vibration monitoring and found that due to low radar power and a long observation distance, random jump errors occurred in the deformation inversion. We proposed an Additive Constant method to solve this problem in [23]. This method adds a constant much larger than the vibration signal to the original radar signal and limits the signal to the right plane of the complex plane, thus avoiding phase wrapping and solving the phase jump problem. However, this method uses the small-amplitude assumption, i.e., that the vibration amplitude is much smaller than the signal wavelength, which limits its application. In addition, this method cannot solve the clutter estimation problem at low SNR.

In this paper, we propose a periodic filtering method to suppress the noise of the original echo signal by using the periodic repetition characteristics of vibration signals. Unlike the low-pass filtering method, which uses a sliding window to average adjacent cells, periodic filtering accumulates signals from several adjacent cycles to suppress noise. This method does not require oversampling for pulse repetition frequency (PRF) and only needs to meet the Nyquist sampling criterion. After periodic filtering, the stationary clutter can be estimated more accurately, and the jump error problem in deformation can also be solved.

The rest of this paper is organized as follows. Section 2 describes the geometry and the signal model of the vibration monitoring radar. Section 3 analyzes the influences of static clutter and noise on deformation inversion. Section 4 introduces the basic principle of periodic filtering, gives the main flow of the algorithm, and derives the expressions of the periodic filter in time and frequency domains. Section 5 verifies the effectiveness and correctness of the proposed method through simulation and real experiments. Section 6 presents the conclusion.

## 2. Geometry and Signal Model of Vibration Monitoring Radar

The observation geometry of the vibration monitoring radar is shown in Figure 1. The radar is fixed. It transmits and receives signals at constant time intervals. The radar beam illuminates the vibration point P. The range from P to the antenna phase center is denoted as $R_0$.

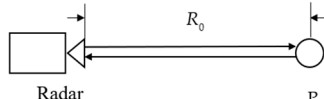

**Figure 1.** Geometry of vibration monitoring radar.

In this paper, the radar system uses Frequency Modulated Continuous Wave (FMCW). The transmitted signal is expressed as follows:

$$s(t) = \exp\left\{ j2\pi f_{\mathrm{c}} t + j\pi K t^2 \right\} \tag{1}$$

where $t$ is the fast time, $f_{\mathrm{c}}$ is the center frequency, and $K$ is the signal frequency modulation rate.

The received signal is a delayed version of the transmitted signal. The transmitted and received signals are then mixed, generating the baseband signal:

$$
\begin{aligned}
s_{\mathrm{v}}(t, v) &= \sigma_{\mathrm{p}} \cdot s(t - t_0) \cdot s^*(t) \\
&= \sigma_{\mathrm{p}} \cdot \exp\left\{ -j2\pi (f_{\mathrm{c}} + Kt) t_0 + j\pi K t_0^2 \right\}
\end{aligned}
\tag{2}
$$

where $v$ is the slow time, $\sigma_{\mathrm{p}}$ is the complex reflectivity of the vibration point P, and $t_0$ is the time delay of the signal from transmission to being reflected by the vibration point P and finally being received by the radar receiver, which can be expressed as follows:

$$t_0 = \frac{2(R_0 + X(v))}{C} \tag{3}$$

where $C$ is the speed of light and $X$ represents the simple harmonic vibration, which can be expressed as a function of slow time:

$$X(v) = A\cos(2\pi f_{\mathrm{v}} v + \varphi_0) \tag{4}$$

where $A$ is the vibration amplitude, $f_{\mathrm{v}}$ is the vibration frequency, and $\varphi_0$ is the initial phase of the simple harmonic vibration.

In Equation (2), the quadratic phase term is called the residual video phase (RVP). It is small, and vibration measurement applications can safely ignore it without significant impact on deformation inversion. Substitute Equation (3) into Equation (2) and define a new frequency according to the linear time-frequency relationship of FMCW as follows:

$$f_r = Kt. \tag{5}$$

Then, Equation (2) can be expressed as follows:

$$s_v(f_r, v) = \sigma_p \cdot \exp\left\{-j\frac{4\pi(f_c + f_r)}{C}(R_0 + X(v))\right\}. \tag{6}$$

An inverse Fourier transform with respect to $f_r$ is performed on Equation (6) to obtain the range compressed signal, which can be expressed as follows:

$$s_v(\tau, v) = \sigma_p \cdot \exp\left\{-j\frac{4\pi f_c}{C}(R_0 + X(v))\right\} \cdot \operatorname{sinc}\left(B_r\left(\tau - \frac{2(R_0 + X(v))}{C}\right)\right) \tag{7}$$

where $\tau$ is the time corresponding to frequency $f_r$ and $B_r$ is the signal bandwidth.

The peak value of the range compressed signal is located at range $R_0$. Vibration information is contained in the one-dimensional signal at distance $R_0$, which changes with slow time $v$. The one-dimensional signal can be expressed as follows:

$$\begin{aligned}
s_v(v) = s_v\left(\tau = \tfrac{2R_0}{C}, v\right) &= \sigma_p \cdot \exp\left\{-j\tfrac{4\pi f_c}{C}(R_0 + X(v))\right\} \\
&= \sigma_p \cdot \exp\left\{-j\tfrac{4\pi}{\lambda_c}(R_0 + X(v))\right\}
\end{aligned} \tag{8}$$

where $\lambda_c = C/f_c$ is the wavelength. Equation (8) is the vibration signal model under ideal conditions. To simplify the following analysis, the phase of $s_v(v)$ is represented by $\phi$, which can be expressed as follows:

$$\phi = -\frac{4\pi}{\lambda_c}(R_0 + X(v)). \tag{9}$$

## 3. Influence of Static Clutter and Noise on Vibration Radar Signals

The real vibration signal contains the clutter from the environment and the noise generated by the system. This paper discusses the influence of static clutter and Gaussian white noise on vibration signals. The vibration signal containing static clutter and Gaussian white noise can be expressed as follows:

$$s(v) = s_v(v) + s_c + s_n \tag{10}$$

where $s_c$ represents the static clutter in the same range cell with the vibrating target, is a complex constant that does not change with the slow time, and $s_n$ represents Gaussian white noise.

In this paper, the static clutter signal model is used. In vibration monitoring, the position of the radar is fixed, and the static clutter signal model can be used when the background is stationary.

There are mainly two situations when the background is not stationary: (1) there are strong targets, such as vehicles or pedestrians passing by, causing fast-changing clutter; and (2) radar position or radar transmitted signal change slowly, resulting in slowly changing clutter.

Fast-changing clutter has a high Doppler frequency, while slowly-changing clutter has a Doppler frequency close to zero. Both types of non-static clutter can be suppressed in the doppler domain range because they have different doppler frequencies from vibration signals. The proposed periodic filter has a comb-shaped spectrum and can filter out non-static clutter, as shown in Section 4.2.

In the following section, the characteristics of the vibration signal and the influence of static clutter and noise on the vibration signal will be analyzed in the complex plane.

### 3.1. Vibration Signal in a Complex Plane

As shown in Figure 2a, an ideal vibration signal is a phase-modulated signal. The real part and the imaginary part form a circle or an arc on the complex plane, with the center at the origin. The radius of the circle or arc is the reflectivity of the vibrating point, that is, the absolute value of the complex reflectivity $|\sigma_p|$, and the phase is the vibration signal phase $\phi$. According to the vibration signal phase shown in Equation (9), when the amplitude $A \geq \lambda/4$, the vibration signal forms a complete circle in the complex plane. Conversely, when the amplitude $A < \lambda/4$, the vibration signal forms an arc in the complex plane.

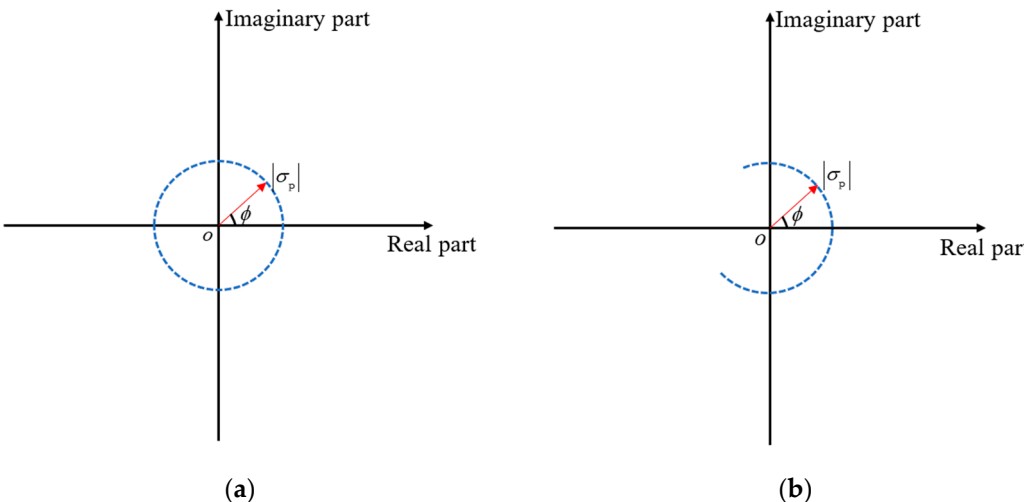

(**a**)            (**b**)

**Figure 2.** Vibration signal in complex plane: (**a**) $|A| \geq \lambda/4$ and (**b**) $|A| < \lambda/4$.

### 3.2. Influence of Static Clutter on Vibration Signal

As shown in Figure 3, in the complex plane, the presence of static clutter makes the center of the circle or arc deviate from the origin. The horizontal and vertical coordinates of the new center are the real part and the imaginary part of the static clutter, respectively. The presence of static clutter will make the signal phase difference smaller than the actual vibration phase difference, resulting in an underestimation of the vibration amplitude. In this figure, $\Delta\phi$ is the phase difference of the signal, and $\Delta\phi_{\text{true}}$ is the actual vibration phase difference. The circle fitting method is commonly used to estimate the center of the complex plane signal so that the static clutter can be estimated and removed.

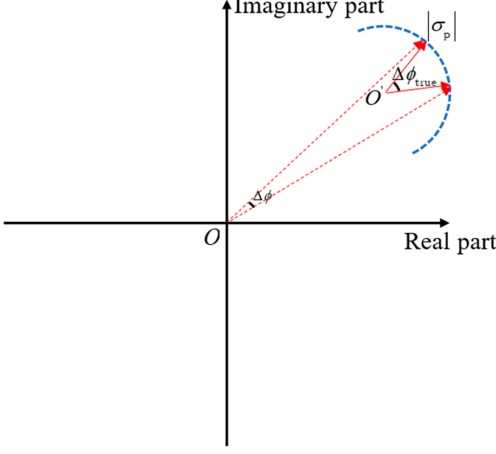

**Figure 3.** Influence of static clutter on vibration signals.

### 3.3. Influence of Noise on the Vibration Signal

When the SNR is low, the strong Gaussian white noise makes the center estimation inaccurate, especially when the arc is only a small portion of a circle, as shown in Figure 4a.

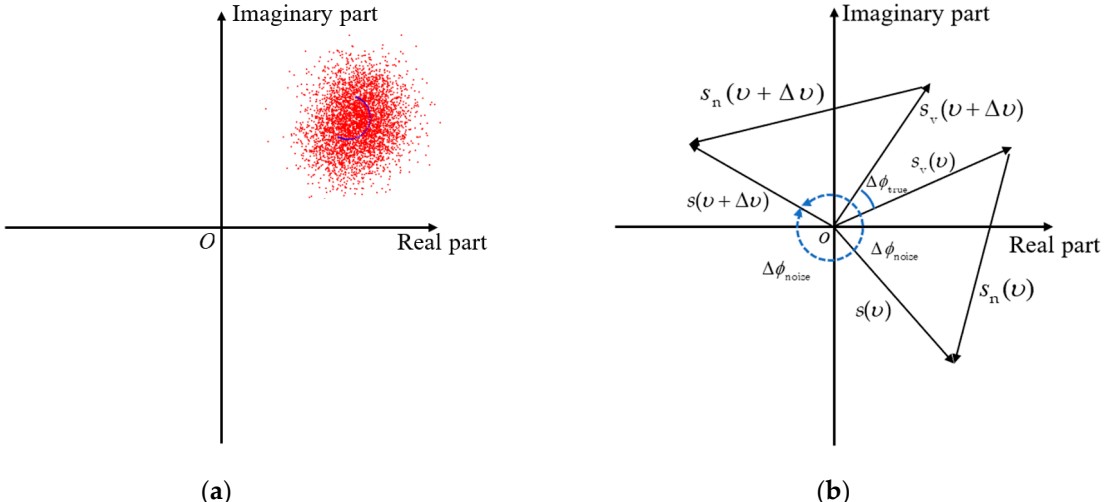

**(a)**                                                                                                  **(b)**

**Figure 4.** Influence of noise on clutter estimation and deformation inversion: (**a**) influence of noise on clutter estimation; and (**b**) influence of noise on deformation inversion.

In addition, under the condition of low SNR, there will be a jump error in deformation inversion. Because of strong noise, the differential phase would be wrapped, and the ambiguity number is unknown. As shown in Figure 4b, the true differential phase is $\Delta\phi_\text{true}$. With the influence of noise, the differential phase is $\Delta\phi_\text{noise}$. The differential phase is wrapped in the range $[-\pi, \pi]$. Because of the randomness of the noise, it is difficult to solve this wrapping problem; that is, the phase ambiguity number is unknown, which will lead to phase jump errors.

In this paper, we define it as a "low SNR" condition when the jump error occurs. Vibration frequency extraction has a low demand for SNR because the signal is accumulated in the Doppler domain. However, the radar signal phase is quite sensitive to noise, and the deformation extraction has a relatively high demand for SNR.

## 4. Clutter Suppression and Deformation Inversion Based on Periodic Filtering

To reduce the influence of noise on clutter estimation and deformation inversion, the traditional method is to perform low-pass filtering on the original vibration radar signal shown in Equation (10). However, the low-pass filtering method requires a high PRF. In the Doppler domain range, the vibration signal spectrum contains multiple harmonics with the target vibration frequency as the base frequency. Therefore, the Doppler bandwidth is usually several times the target vibration frequency. The bandwidth of the low-pass filter needs to be greater than the Doppler bandwidth; otherwise, the Doppler spectrum would be damaged, resulting in signal distortion and inaccurate deformation inversion. In order to achieve a good filtering effect, the PRF needs to be several times the bandwidth of the low-pass filter. When PRF is limited, it is difficult to achieve a good denoising effect without causing signal distortion.

This section will first derive the range Doppler domain expression of the vibration signal and give the Doppler bandwidth expression. Then, the basic principle of periodic filtering is introduced, and the expressions of periodic filtering in the time domain and frequency domain are derived. Next, the circle-fitting clutter estimation method and the deformation inversion method are briefly introduced. Finally, the algorithm flow of this paper is given.

### 4.1. Doppler Domain Vibration Signal

The Doppler domain signal is the Fourier transform of the one-dimensional slow time signal shown in Equation (8). The Doppler spectrum reflects the frequency characteristics of the vibrating target. Because the vibration signal is periodic, its spectrum is discrete, and the frequency components include the vibration frequency and its harmonic frequencies. The Doppler domain signal of a vibrating target can be expressed as follows:

$$
\begin{aligned}
s_{\mathrm{d}}(f_{\mathrm{d}}) &= \int_v s_{\mathrm{v}}(v) \exp\{-j2\pi f_{\mathrm{v}} v\} dv \\
&= \sigma_{\mathrm{p}} \cdot \exp\left\{-j\tfrac{4\pi}{\lambda_{\mathrm{c}}} R_0\right\} \int_v \exp\left\{-j\tfrac{4\pi}{\lambda_{\mathrm{c}}} X(v) - j2\pi f_{\mathrm{v}} v\right\} dv
\end{aligned}
\tag{11}
$$

Since the Doppler spectrum is discrete, only the spectrum at the harmonic frequency $f_{\mathrm{d}} = m \cdot f_{\mathrm{v}}$ need to be calculated, where $m = 0, 1, \ldots$, and only the signal within one cycle needs to be integrated. The expression is as follows:

$$
\begin{aligned}
s_{\mathrm{d}}(m) &= \sigma_{\mathrm{p}} \cdot \exp\left\{-j\tfrac{4\pi}{\lambda_{\mathrm{c}}} R_0\right\} \int_{v=0}^{1/f_{\mathrm{p}}} \exp\left\{-j\tfrac{4\pi A}{\lambda_{\mathrm{c}}} \cos(2\pi f_{\mathrm{v}} v) - j2\pi m f_{\mathrm{v}} v\right\} dv \\
&= \sigma_{\mathrm{p}} \cdot \exp\left\{-j\tfrac{4\pi}{\lambda_{\mathrm{c}}} R_0\right\} \cdot J_m\left(\tfrac{4\pi A}{\lambda_{\mathrm{c}}}\right)
\end{aligned}
\tag{12}
$$

where $J_m$ is the Bessel function of order $m$.

In the Doppler domain, the energy of the periodic signal is accumulated at the harmonic frequency, so the SNR of the spectrum is very high. The base frequency $f_{\mathrm{v}}$ can be easily obtained by peak detection of the spectrum, which provides the input parameter for the design of the periodic filter.

The Doppler bandwidth can be calculated using the phase modulation property of the vibration radar signal. The instantaneous Doppler frequency can be obtained by the derivative of the signal phase shown in Equation (9) with respect to slow time, which can be expressed as follows:

$$
f_{\mathrm{d}} = \frac{1}{2\pi} \cdot \frac{d\phi}{dv} = \frac{-\tfrac{2}{\lambda_{\mathrm{c}}}(R_0 + x(v))}{dv} = \frac{4\pi A f_{\mathrm{v}}}{\lambda_{\mathrm{c}}} \sin(2\pi f_{\mathrm{v}} v + \varphi_0)
\tag{13}
$$

When $\sin(2\pi f_{\mathrm{v}} v + \varphi_0) = 1$, the Doppler frequency reaches the maximum, that is:

$$
(f_{\mathrm{d}})_{\max} = \frac{4\pi A f_{\mathrm{v}}}{\lambda_{\mathrm{c}}}.
\tag{14}
$$

When $A > \lambda_{\mathrm{c}}/(4\pi)$, the Doppler bandwidth of the echo signal is several times the target vibration frequency. According to the Nyquist sampling criteria, the PRF should be greater than twice the maximum Doppler frequency, that is:

$$
\mathrm{PRF} \geq 2(f_{\mathrm{d}})_{\max}.
\tag{15}
$$

### 4.2. Periodic Filtering Method

The periodic filtering method proposed in this paper makes use of the periodic repetition characteristics of vibration signals to accumulate the signals of several adjacent cycles to suppress noise. Compared with the low-pass filtering method, this method has a low requirement for PRF and only needs to meet the Nyquist sampling criteria.

The expression of time-domain periodic filtering is:

$$
s_{\mathrm{f}}(v) = \int_{t=0}^{+\infty} s(t) g(v - t) dt = s(v) \otimes g(v)
\tag{16}
$$

where symbol $\otimes$ represents convolution and $g(v)$ is the periodic filter, which can be expressed as follows:

$$g(v) = \frac{\delta(v) + \sum\limits_{m=1}^{M} (\delta(v - mT) + \delta(v + mT))}{1 + 2M} \tag{17}$$

where $\delta$ represents the impulse function, $T = 1/f_v$ is the vibration period, $m = 1, 2, \ldots M$ is the order number of adjacent cycles involved in filtering, and $M$ is manually set.

The physical meaning of Equation (16) is that the signals of adjacent $1 + 2M$ cycles are averaged, and the SNR is increased by $1 + 2M$ times. The greater the $M$, the better the noise suppression effect.

Time-domain convolution is equivalent to frequency domain multiplication. The frequency-domain expression of the periodic filter can be expressed as follows:

$$
\begin{aligned}
G(f_d) &= \int\limits_{v=0}^{+\infty} g(v) \cdot \exp(-j2\pi f_d v) dv \\
&= \frac{1 + \sum\limits_{m=1}^{M} (\exp(-j2\pi m f_d T) + \exp(j2\pi m f_d T))}{1 + 2M} \\
&= \frac{1 + 2 \sum\limits_{m=1}^{M} \cos(2\pi m f_d T)}{1 + 2M}
\end{aligned} \tag{18}
$$

The expression of frequency-domain periodic filtering can be expressed as follows:

$$s_f(v) = \frac{1}{2\pi} \int\limits_{f_d=-\infty}^{+\infty} s_d(f_d) \cdot G(f_d) \exp\{j2\pi f_d v\} df_d. \tag{19}$$

The figures of the frequency domain low-pass filter and periodic filter are shown in Figure 5. Since the filter is symmetrical about zero Doppler, only the positive axis of Doppler frequency is shown in the figure. The low-pass filter is a rectangular window, and the periodic filter is a group of narrow pulses with $f_v$ as the interval. The periodic filter has a comb-shaped spectrum; the larger the $M$, the narrower the pulses, and the better the filtering effect. The comb-shaped filter can filter out noise and non-static clutter.

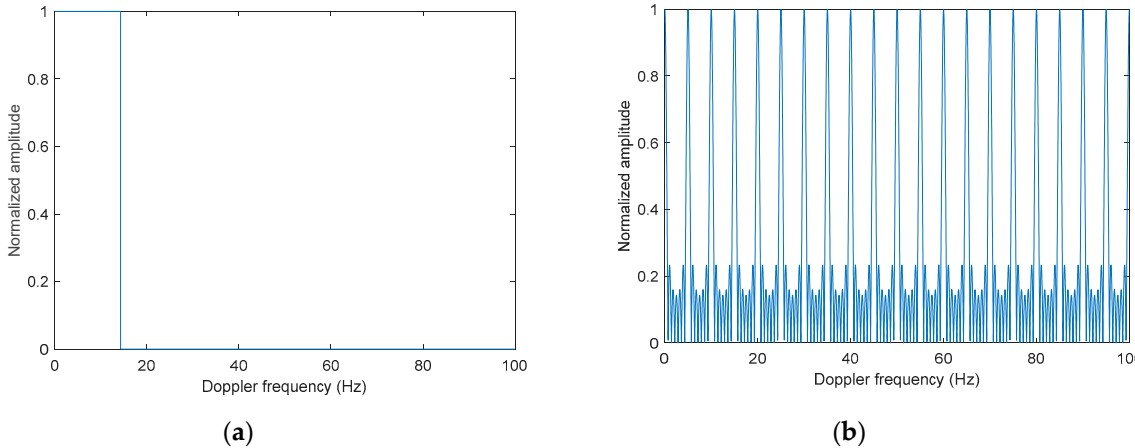

**Figure 5.** Frequency domain filters: (**a**) low-pass filter and (**b**) periodic filter.

Two main factors should be considered when selecting the value of $M$. The first factor is SNR. The $M$ value needs to be large enough to avoid jumping errors in deformation inversion. The second factor is the time-variation characteristics of the vibration signal. If

the vibration frequency, vibration amplitude, and clutter stay constant or change slowly with time, the value of $M$ can be larger, and vice versa.

In this paper, we make use of the periodic repetition of a vibration signal and do not use the small vibration amplitude assumption. For both cases, the proposed periodic filtering method is effective whether the vibration amplitude is greater than or less than a quarter of the wavelength.

### 4.3. Circle-Fitting Clutter Estimation

After periodic filtering, the signal SNR increases. The next step is to estimate the clutter. The least squares circle fitting method is usually used to estimate the center and radius of the complex plane vibration signal [21]. The center of the circle represents the clutter, and the radius represents the scattering coefficient of the vibrating target. Let the center of the circle be $x$, $x = (I(s_c), Q(s_c))^T$, the radius be $r$, $r = |\sigma_p|$, and the vibration signal be $s_k = (I(s_f(v_k)), Q(s_f(v_k)))^T$, where $I$ denotes the real part of the signal, $Q$ denotes the imaginary part, the symbol T represents matrix conjugate transpose, $v_k$ is the discrete value of the slow time, and $k = 1, 2, \ldots$. The least square circle fitting is to minimize the following cost function:

$$\min \sum_k (g_k(x, r))^2 \tag{20}$$

where

$$\begin{aligned} g_k(x, r) &= \|s_k - x\|_2^2 - r^2 \\ &= (s_k - x)^T (s_k - x) - r^2 \\ &= s_k^T s_k - (2s_k^T x + r^2 - x^T x) \end{aligned} \tag{21}$$

To express Equation (20) in the form of a matrix operation, let:

$$Y = \left( s_1^T s_2, \ldots, s_k^T s_k, \ldots \right)^T = \left( \|s_1\|_2^2, \ldots, \|s_k\|_2^2, \ldots \right)^T \tag{22}$$

$$\beta = \begin{pmatrix} 2x \\ r^2 - x^T x \end{pmatrix}, \tag{23}$$

additionally,

$$X = \begin{pmatrix} s_1^T, 1 \\ \cdots \\ s_k^T, 1 \\ \cdots \end{pmatrix}. \tag{24}$$

Substitute Equations (21)–(24) into Equation (20), and the cost function can be rewritten as follows:

$$\min \sum_k (g_k(x, r))^2 = \min \|Y - X\beta\|_2^2, \tag{25}$$

then the parameter to be estimated is:

$$\hat{\beta} = \mathrm{argmin}_\beta \|Y - X\beta\|_2^2. \tag{26}$$

The solution of the above equation is:

$$\hat{\beta} = \left( X^T X \right)^{-1} X^T Y. \tag{27}$$

The estimated circle center and radius can be expressed as follows:

$$\begin{cases} \hat{x}_1 = \hat{\beta}_1 / 2 \\ \hat{x}_2 = \hat{\beta}_2 / 2 \\ \hat{r} = \sqrt{\hat{\beta}_3 + \hat{x}_1^2 + \hat{x}_2^2} \end{cases} \tag{28}$$

where $\beta_1$, $\beta_2$, $\beta_3$ are the three elements of vector $\beta$, and $x_1$, $x_2$ are the horizontal and vertical coordinates of the circle center, respectively.

The estimated clutter signal is:

$$\hat{s}_c = x_1 + jx_2. \tag{29}$$

*4.4. Deformation Inversion*

After the static clutter is estimated and removed from the periodic filtered signal, the estimated vibration signal is:

$$\widehat{s}_v = s_f(v) - \widehat{s}_c. \tag{30}$$

Deformation inversion is applied to the estimated vibration signal. To avoid phase wrapping, the differential phase is obtained by interference processing first, and then the deformation is obtained by accumulating the differential phase.

The expression of the differential phase is as follows:

$$\phi_d(v_k) = \arg\left(\widehat{s}_v(v_{k+1}) \cdot \widehat{s}_v^*(v_k)\right) \tag{31}$$

where the function arg is to obtain the phase of a complex value. The deformation has the expression of:

$$\widehat{X}(v_k) = -\frac{\lambda}{4\pi} \sum_{m=1}^{k} \phi_d(v_m). \tag{32}$$

Periodic filtering has removed part of the noise and solved the problem of differential phase jump errors caused by noise. The deformation has a frequency of $f_v$, so its bandwidth is much narrower than that of the original vibration radar signal. In other words, deformation inversion reduces the signal bandwidth. Therefore, the deformation can be further filtered by a low-pass filter, such as a time-domain Butterworth filter or a frequency-domain rectangular filter. Taking frequency domain low-pass filtering as an example, its expression is:

$$\widehat{X}_f(v) = I\left(IFT\left(FT\left(\widehat{X}_f(v)\right) \cdot H_{lf}(f_d)\right)\right) \tag{33}$$

where FT represents the Fourier transform, IFT represents the inverse Fourier transform, and $H_{lf}$ represents the frequency domain low-pass filter, which has the expression of:

$$H_{lf} = \text{rect}\left(\frac{f_d}{B_{lf}}\right) \tag{34}$$

where rect represents a rectangular function and $B_{lf}$ is the bandwidth of the low-pass filter, which needs to be greater than twice the vibration frequency, that is:

$$B_{lf} > 2f_v. \tag{35}$$

*4.5. Algorithm Flow chart*

The main steps of the algorithm are shown in Figure 6, including:

1. Vibration frequency estimation. The vibration frequency $f_v$ is the base frequency of the Doppler signal $s_d(f_d)$. It can be extracted by peak detection, and it is the input parameter for the design of the periodic filter;
2. Periodic filtering. The frequency-domain periodic filter $G(f_d)$ is shown in Equation (18). The signal $s_d(f_d)$ is multiplied with $G(f_d)$ to obtain the filtered signal $s_f(v)$, whose SNR is improved;
3. Clutter estimation. The least squares circle fitting method is used to estimate the clutter of the periodically filtered signal $s_f(v)$, and then the estimated clutter $\hat{s}_c$ is subtracted from $s_f(v)$ to obtain the estimated vibration signal $\widehat{s}_v$;

4. Deformation inversion. The differential phase of $\widehat{s}_\mathrm{v}$ is obtained by interference processing, and then the deformation $\widehat{X}(v_k)$ is obtained by accumulating the differential phase;

5. Low-pass filtering. The deformation $\widehat{X}(v_k)$ has a frequency of $f_\mathrm{v}$, and a bandwidth that is much smaller than the Doppler signal. The noise in the deformation can be further filtered by low-pass filtering.

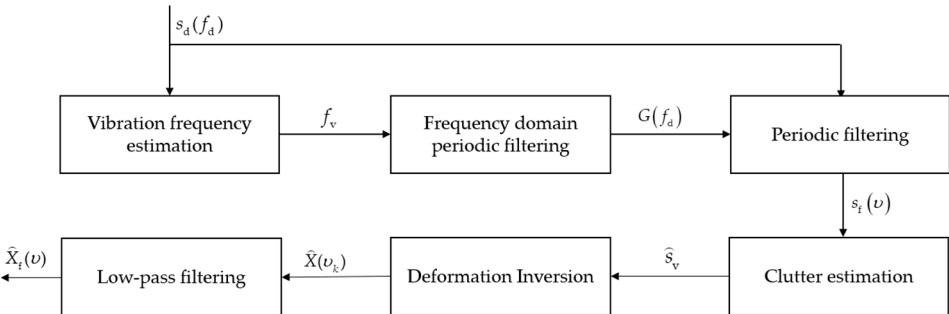

**Figure 6.** Flow chart of clutter estimation and deformation inversion based on periodic filtering.

## 5. Experimental Verification

This section verifies the correctness and effectiveness of the proposed method through simulation experiments, vibrating calibrator experiments, and bridge vibration monitoring experiments. The radar system used in the experiments is the FMCW 77 GHz millimeter wave radar. The experimental parameters are listed in Table 1.

**Table 1.** Radar parameters.

| Parameter | Value |
|:---:|:---:|
| Operating waveform | FMCW |
| Minimum frequency (GHz) | 77 |
| Chirp rate (MHz/μs) | 5.0210 |
| Sampling frequency (MHz) | 5 |
| Samples in chirp | 512 |
| Pulse repetition rate (HZ) | 200 |

### 5.1. Simulation Experiment

In the simulation experiment, the vibration target is 20 m away from the radar. The scattering coefficient of the vibration target is set to one, the vibration amplitude is set to 1 mm, the vibration frequency is set to 5 Hz, the static clutter at the same range bin is set to 10 + 10j, and the SNR is set to 5 dB.

Figure 7 shows the simulation results. Figure 7a–d show the results without periodic filtering. Figure 7a shows the original vibration signal on the complex plane, where the blue dots present the noisy signal and the red dots present the ideal noiseless signal. In the figure, due to the influence of the clutter, the center of the signal deviates from the origin. In addition, due to the influence of the strong noise, the blue dots randomly deviate from the ideal red circle. Without filtering, the clutter estimated by circle fitting is 10.0612 + 9.9416j. Figure 7b shows the Doppler spectrum of the original vibration radar signal, which has multiple harmonic characteristics. The base frequency is equal to the target vibration frequency, and the measured value is about 5 Hz. Figure 7c shows the deformation inversion result after removing the estimated clutter. We can see that strong noise causes serious jump error problems. Figure 7d is the spectrum of the deformation. The frequency spectrum includes not only the target vibration frequency but also other frequency components caused by deformation inversion errors, which are mainly in the low frequency band.

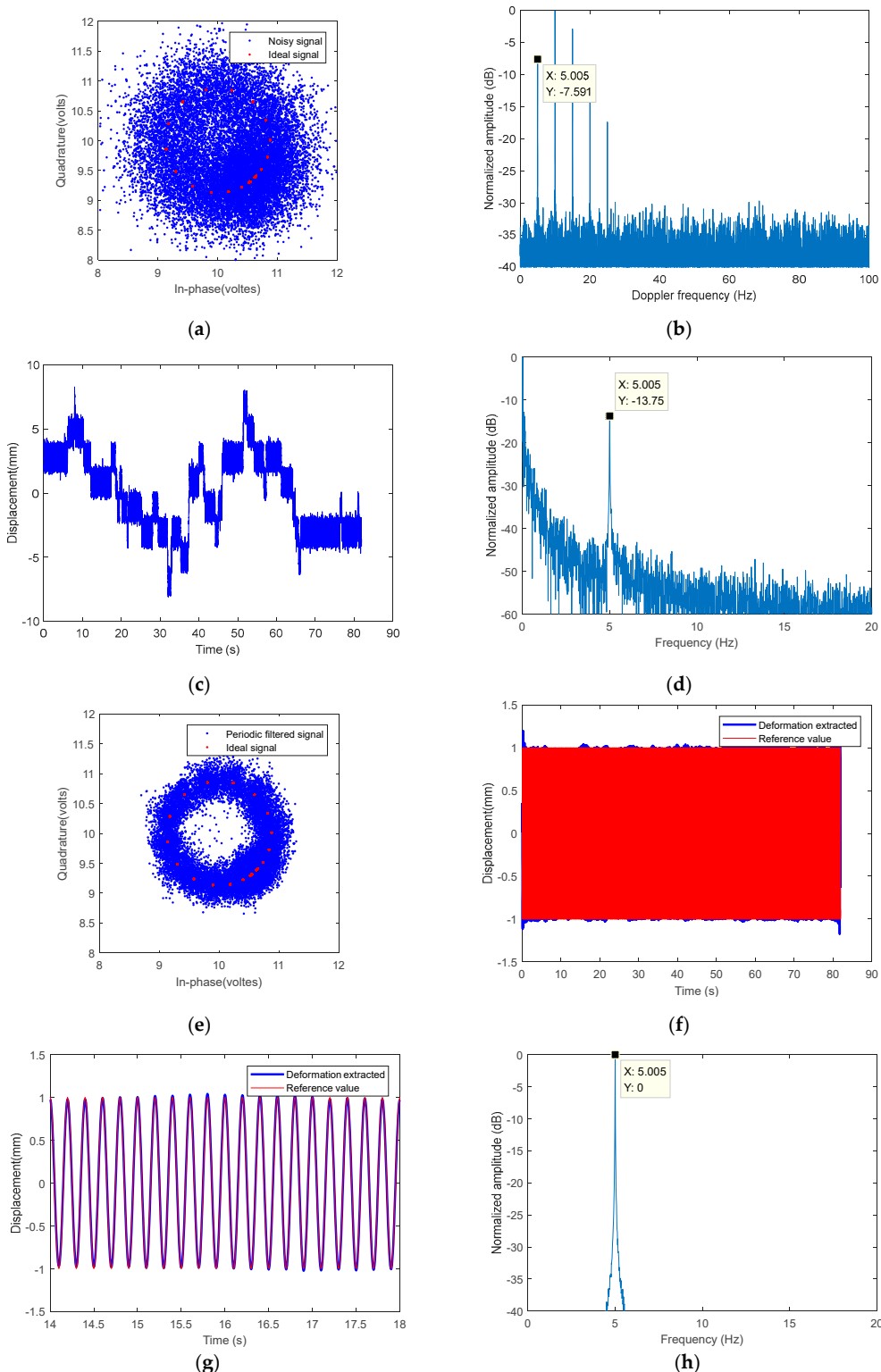

**Figure 7.** Simulation experimental results: (**a**) original complex plane vibration signal; (**b**) original Doppler signal; (**c**) deformation extracted from the original vibration signal; (**d**) spectrum of the deformation extracted from the original vibration signal; (**e**) complex plane vibration signal after periodic filtering; (**f**) deformation extracted after periodic filtering; (**g**) detailed deformation after periodic filtering; and (**h**) frequency spectrum of deformation after periodic filtering.

Figure 7e–h show the results with periodic filtering. In this experiment, the parameter *M* of the periodic filter is set to three, and the SNR can be improved by seven times

theoretically. Figure 7e shows the filtered vibration signals on the complex plane. Compared with the original signal shown in Figure 7e, the noise is significantly suppressed. The estimated clutter is 10.0138 + 9.9917i, which proves that the periodic filtering method can improve the clutter estimation accuracy. Figure 7f shows the result of deformation inversion. Compared with the unfiltered result shown in Figure 7c, the jump error problem caused by noise is solved because of the improved SNR. Figure 7g shows the deformation inversion results in more detail. The deformation extracted from the radar signal is very close to the simulation preset. Figure 7h shows the frequency spectrum of the extracted deformation, which mainly includes the target vibration frequency. The experimental results show that the periodic filtering method can estimate static clutter more accurately and solve the problem of jump errors in deformation inversion.

In order to compare the proposed periodic filtering method with the traditional low-pass filtering method, simulation experiments are carried out under different SNRs. From the Doppler signal shown in Figure 7b, we can see that the data is oversampled. In the following simulation, the PRF is set to 100 Hz to analyze the performance of the two methods when the data is not oversampled. The results are shown in Figures 8–10, and the SNR is set to 10 dB, 5 dB, and 0 dB, respectively.

Figure 8 shows the results under 10 dB SNR. As shown in Figure 8b, there is a jump error in the deformation at the time of about 96 s. Figure 8c shows the low-pass filtered signal in the complex plane. The bandwidth of the low-pass filter is about 33.33 Hz, one-third of the PRF, which can increase the SNR by three times. It can be seen from Figure 7b that the main energy of the vibration signal spectrum is within 25 Hz. Since the filter bandwidth is less than twice the signal bandwidth, the filtered signal will have a little distortion. Figure 8d shows the deformation extracted after low-pass filtering. The extracted deformation is not a pure sine wave, but a little distorted. Figure 8e,f are the results of the proposed periodic filtering method. The $M$ value of the filter is set to one to increase the SNR by three times. From the results, we can see the noise is effectively suppressed, and the extracted deformation is very close to the reference value.

Figure 9 shows the results under 5 dB SNR. As shown in Figure 9b, the jump error problem is very serious due to the relatively low SNR. Figure 9c shows the low-pass filtered signal in the complex plane. The bandwidth of the low-pass filter is 20 Hz, one fifth of the PRF, which can increase the SNR by five times. Since the filter bandwidth is much less than twice the signal bandwidth, the signal is distorted. As shown in Figure 9d, the deformation cannot be correctly extracted. Figure 9e,f are the results of the proposed periodic filtering method. The $M$ value of the filter is set to two to increase the SNR by five times. From the results, we can see the noise is effectively suppressed, and the extracted deformation is still very close to the reference value.

Figure 10 shows the results under 0 dB SNR. Because of the low SNR, the problem of jump errors in deformation inversion is more serious. In this experiment, the bandwidth of the low-pass filter is about 11.11 Hz, which is one-ninth of the PRF. Due to information loss caused by low-pass filtering, deformation cannot be correctly extracted. The $M$ value of the periodic filter is set to four to increase the SNR by nine times. Under conditions of very low SNR, the proposed periodic filtering method can still effectively suppress the noise and accurately extract the deformation.

Table 2 compares the clutter estimation results with and without periodic filtering. It shows that the estimation error of methods with periodic filtering is significantly smaller than that without periodic filtering.

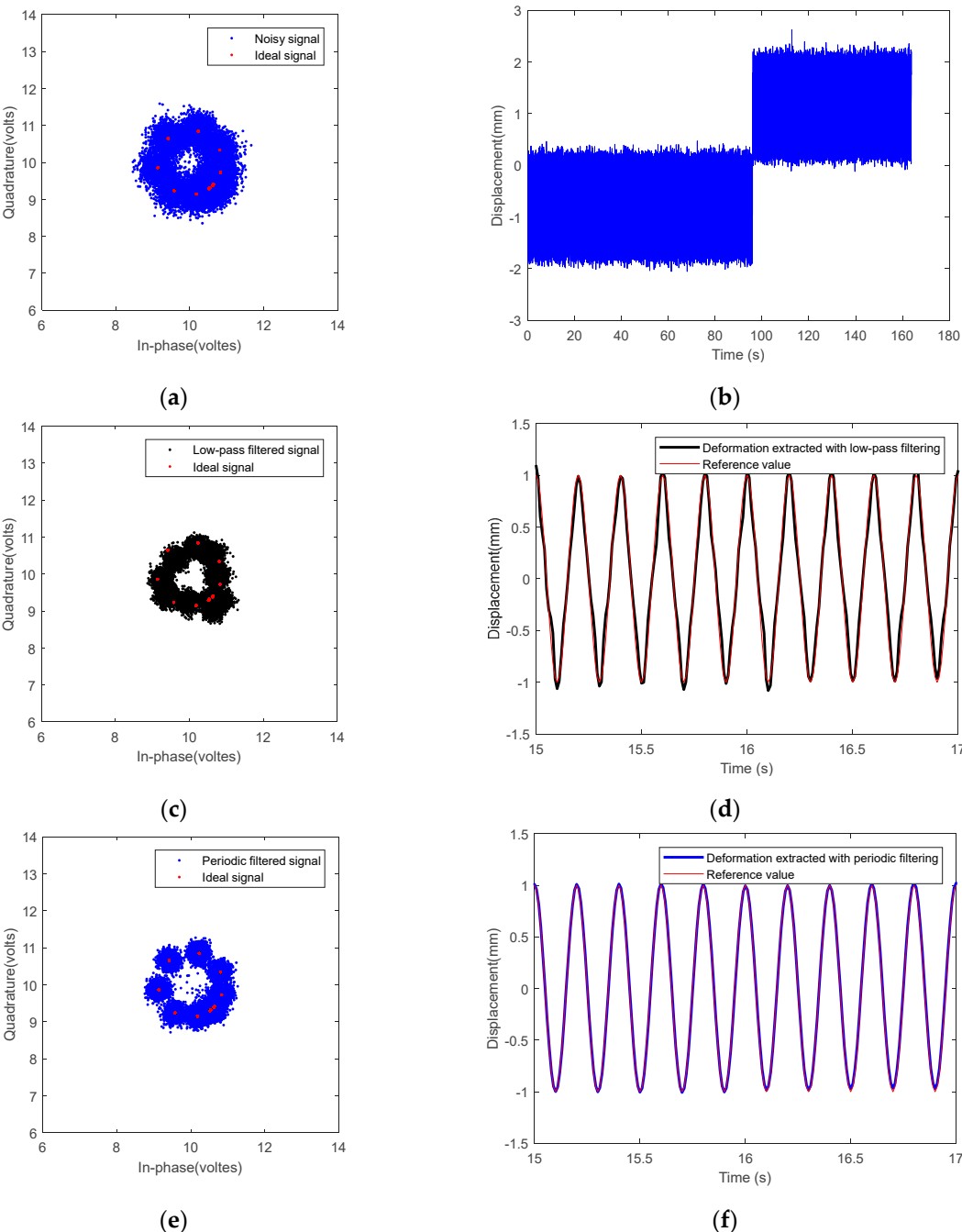

**Figure 8.** Comparison of low-pass filtering and periodic filtering under 10 dB SNR: (**a**) original complex plane vibration signal; (**b**) deformation extracted from the original vibration signal; (**c**) complex plane vibration signal after low-pass filtering (the bandwidth is one third of the PRF); (**d**) deformation extracted after low-pass filtering; (**e**) complex plane vibration signal after periodic filtering (*M* = 1); and (**f**) deformation extracted after periodic filtering.

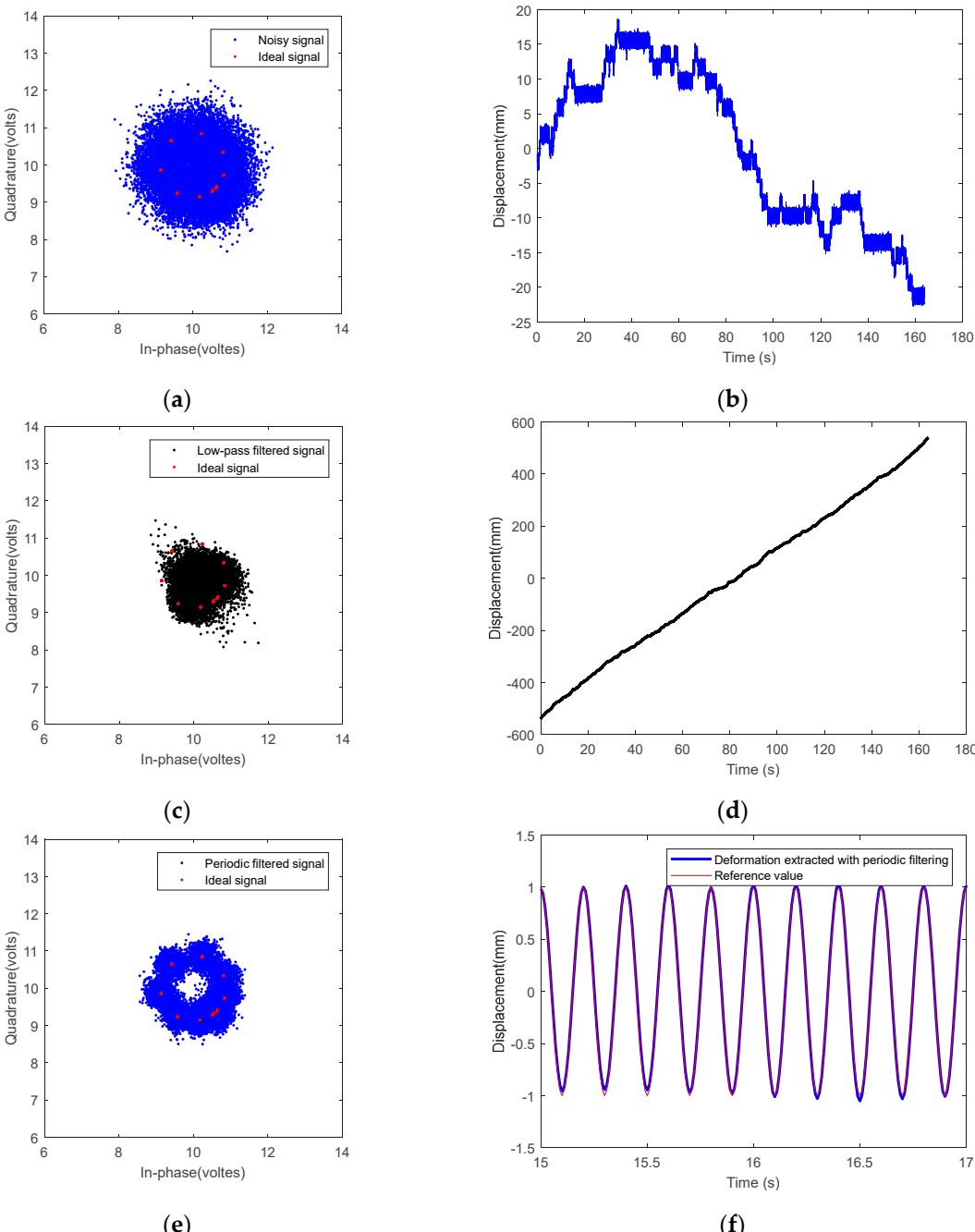

**Figure 9.** Comparison of low-pass filtering and periodic filtering under 5 dB SNR: (**a**) original complex plane vibration signal; (**b**) deformation extracted from the original vibration signal; (**c**) complex plane vibration signal after low-pass filtering (the bandwidth is one fifth of the PRF); (**d**) deformation extracted after low-pass filtering; (**e**) complex plane vibration signal after periodic filtering (*M* = 2); and (**f**) deformation extracted after periodic filtering.

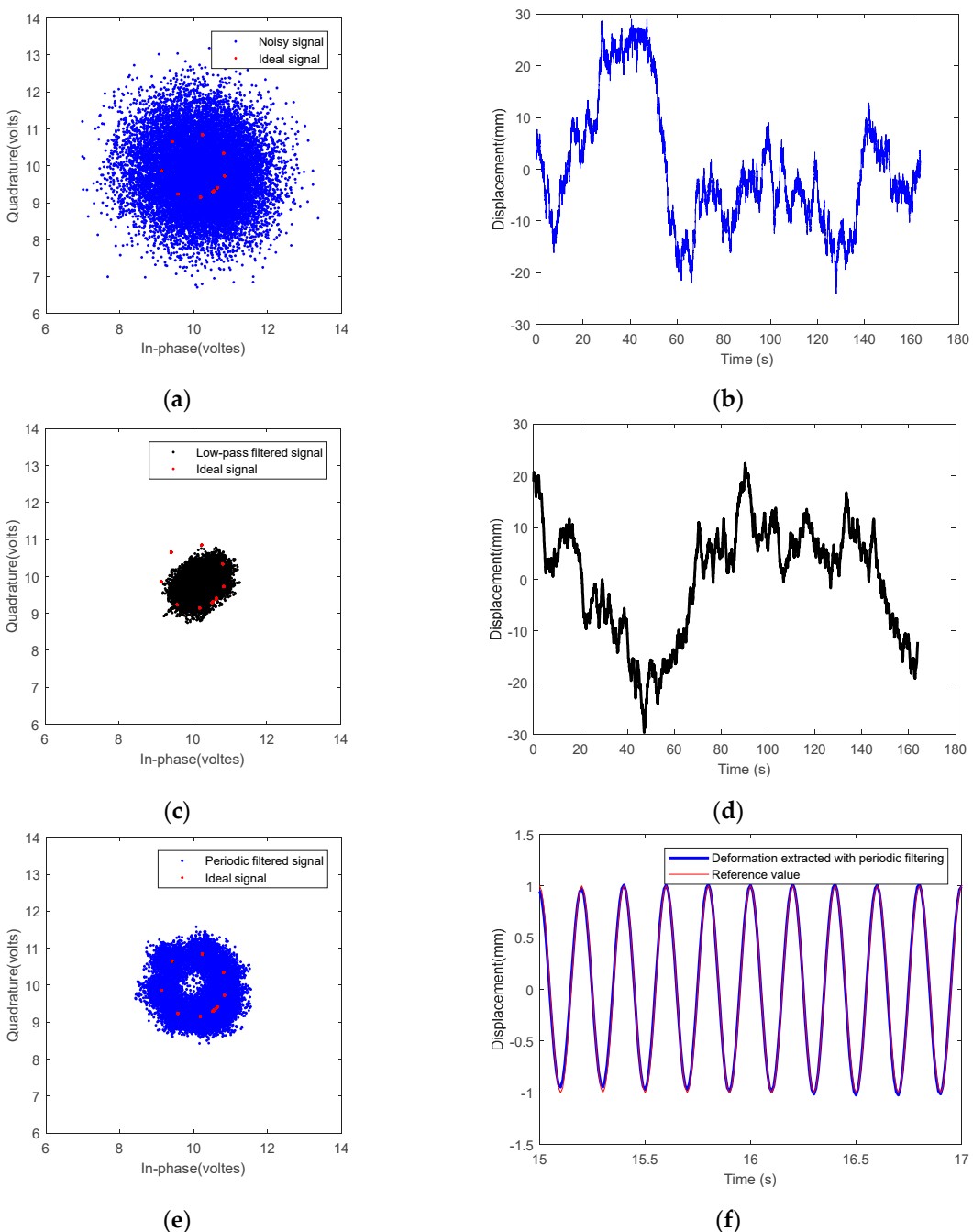

**Figure 10.** Comparison of low-pass filtering and periodic filtering under 0 dB SNR: (**a**) original complex plane vibration signal; (**b**) deformation extracted from the original vibration signal; (**c**) complex plane vibration signal after low-pass filtering (the bandwidth is one ninth of the PRF); (**d**) deformation extracted after low-pass filtering; (**e**) complex plane vibration signal after periodic filtering (*M* = 4); and (**f**) deformation extracted after periodic filtering.

**Table 2.** Clutter estimation results.

| SNR | Clutter Estimation Error without Periodic Filtering | Clutter Estimation Error with Periodic Filtering |
|---|---|---|
| 10 dB | 0.0219–0.0268i | 0.0082–0.0121i |
| 5 dB | 0.0521–0.0591i | 0.0179–0.0115i |
| 0 dB | 0.1251–0.1185i | 0.0398–0.0304i |

This paper mainly validates the proposed periodic filtering for solving the jump error problem. A controlled experiment on clutter depression can be referred to [24].

### 5.2. Vibrating Calibrator Experiment

The radar we used for real-time data validation is the TI IWR1642 BoosterPack. It has two transmit antennas and four receive antennas, but we only used data with one transmit antenna and one receive antenna. The raw samples are captured through a TI DCA1000EVM data acquisition board. The parameters are the same as the simulation parameters listed in Table 1.

In this experiment, the target is a vibrating calibrator, which can vibrate at a known frequency and amplitude. Figure 11a shows a photo of the vibrating calibrator. The calibrator is a vertically placed metal plate that is driven by a motor and vibrates along a track with a certain frequency and amplitude. The calibration device uses a high-precision screw and a motor. The positioning accuracy of the screw is 0.05 mm, the screw moves 10 mm in one revolution, and the maximum operating speed is 50 mm/s. The motor produces 6400 pulses/rev. In the experiment, we set the one-way movement of the calibrator to 1 mm and the reciprocating frequency to 2 Hz. This is far less than the maximum operating speed limit of the screw, and the positioning accuracy of the screw meets the requirements of a submillimeter deformation measurement. At the same time, the motor pulse is generated by an FPGA with a working frequency of 100 MHz. The FPGA generates 640 pulses every 0.25 s to complete the movement of 1 mm, ensuring high precision of the frequency. The experimental scene is shown in Figure 11b. The vibrating calibrator is placed about 60 m away from the radar, and a corner reflector is placed beside it to create strong static clutter at the same range bin.

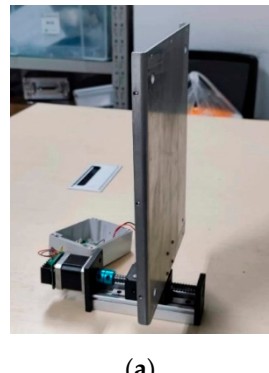

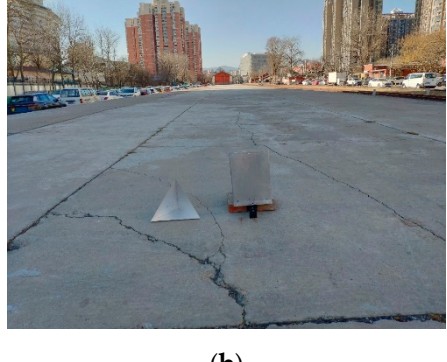

(**a**)　　　　　　　　　　　　　　　　　(**b**)

**Figure 11.** Photo of the vibrating calibrator and the experimental scene: (**a**) vibrating calibrator and (**b**) experimental scene.

The experiment results for the vibrating calibrator are shown in Figure 12. Figure 12a–d show the unfiltered results. Figure 12a shows the original vibration signal in the complex plane, and the estimated clutter is 615.40 + 23.450j. Figure 12b shows the spectrum of the original vibration signal. The detected base frequency is about 2 Hz, which is consistent with the reference value. Figure 12c shows the deformation extracted from the original vibration signal. The jump error problem caused by strong noise is very serious. Figure 12d shows the spectrum of the extracted deformation. Due to the serious jump error, the energy of the vibration frequency component is very weak.

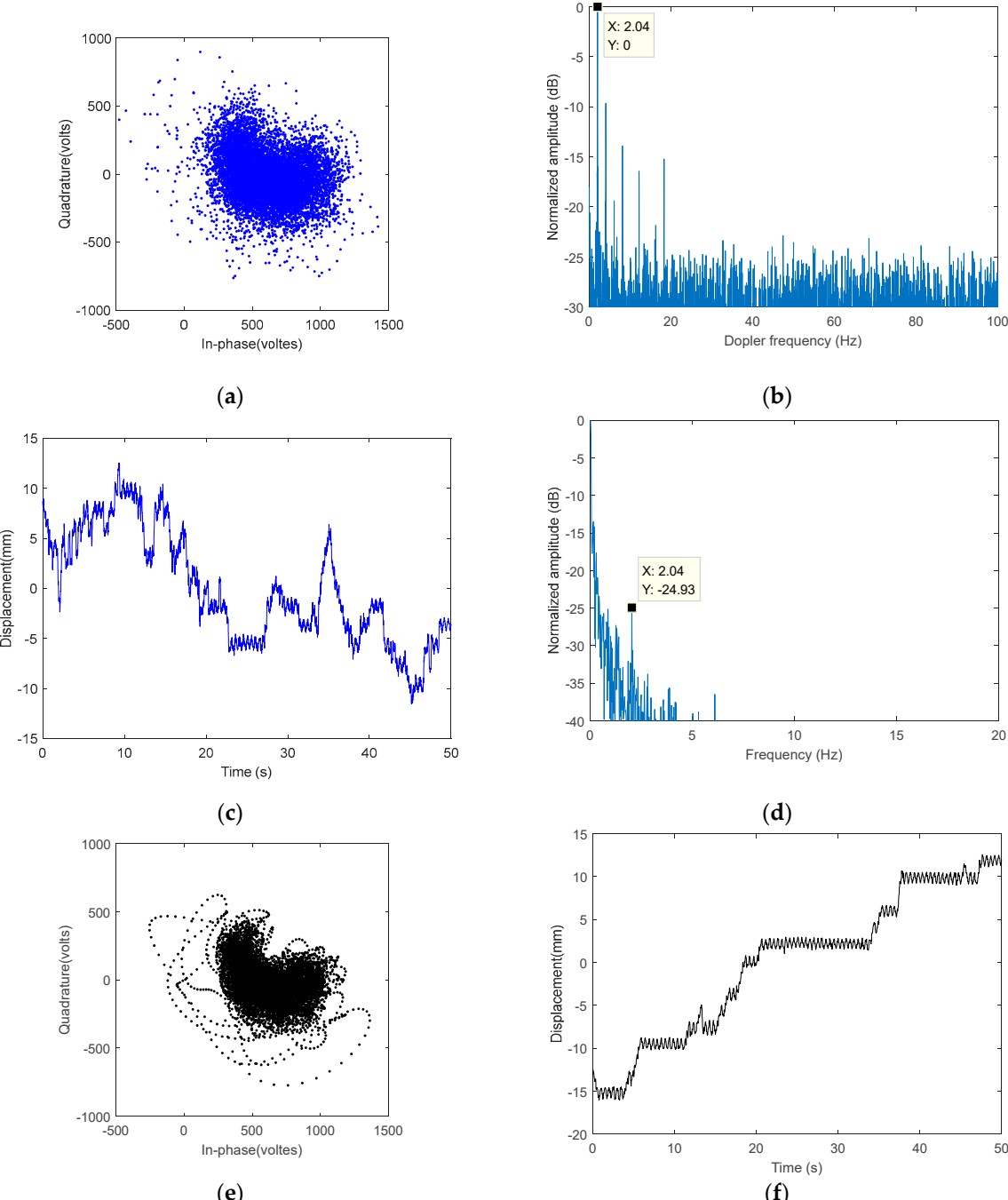

**Figure 12.** *Cont.*

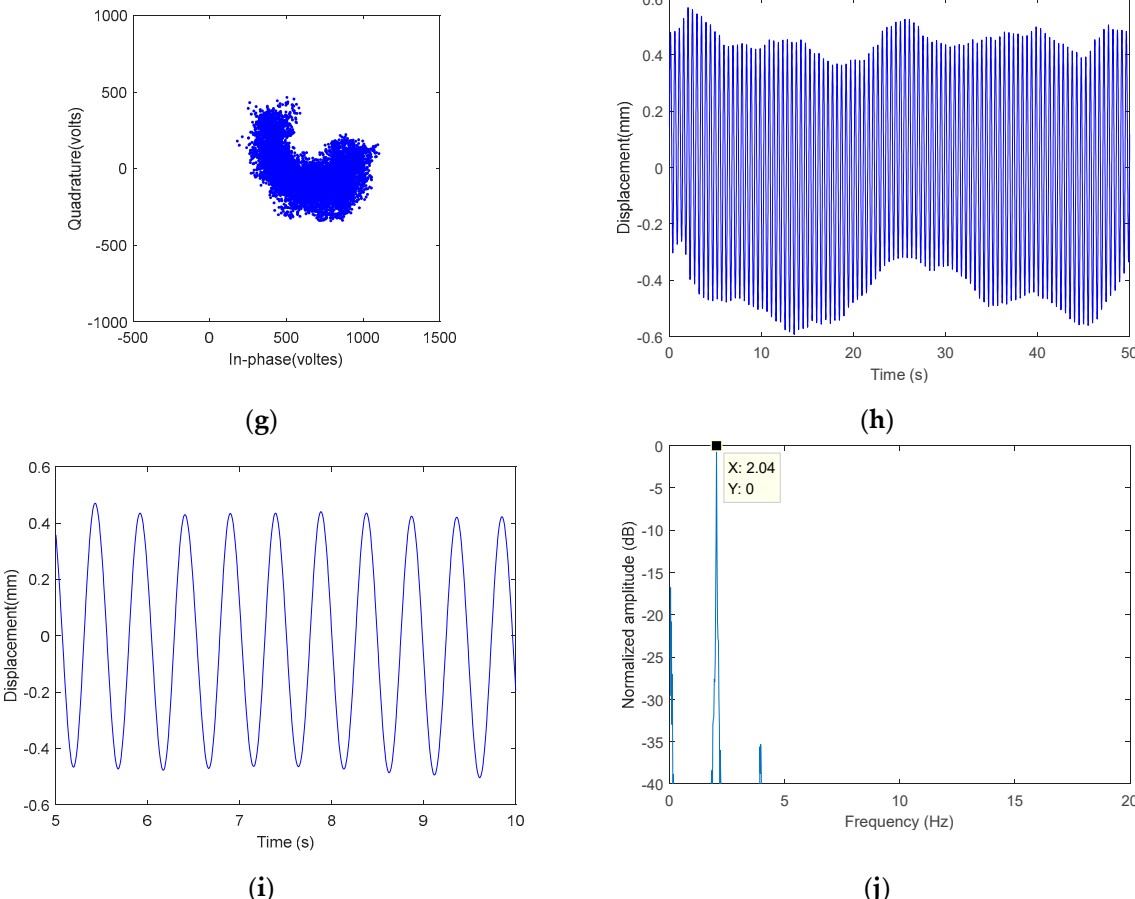

**Figure 12.** Vibrating calibrator experiment results: (**a**) original complex plane vibration signal; (**b**) original Doppler signal; (**c**) deformation extracted from the original vibration signal; (**d**) spectrum of the deformation extracted from the original vibration signal; (**e**) complex plane vibration signal after low-pass filtering (the bandwidth is one ninth of the PRF); (**f**) deformation extracted after low-pass filtering; (**g**) complex plane vibration signal after periodic filtering ($M = 4$); (**h**) deformation extracted after periodic filtering; (**i**) detailed deformation after periodic filtering; and (**j**) frequency spectrum of deformation after periodic filtering.

Figure 12e,f show the results of the traditional low-pass filtering method. The bandwidth of the low-pass filter is set to 11.11 Hz, which is one-ninth of the PRF. It can be seen from Figure 12b that the main energy of the vibration signal spectrum is within 20 Hz. Since the filter bandwidth is less than twice the signal bandwidth, some information in the signal will be lost. Therefore, the deformation cannot be extracted correctly.

Figure 12g–j show the results with periodic filtering. Because the SNR in this experiment is low, $M$ in the periodic filter is set to four, so the SNR can theoretically be improved by nine times. Figure 12g shows the complex plane vibration signal after periodic filtering. Compared with the original signal shown in Figure 12a, the noise is significantly suppressed after periodic filtering, and the estimated clutter is 664.66 + 87.84j. Figure 12h shows the extracted deformation after periodic filtering. With the increase in SNR, the problem of jump errors is solved. Figure 12i shows the deformation in detail, and the vibration amplitude is about 0.5 mm, which is consistent with the reference value. Figure 12j shows the deformation spectrum, and the vibration frequency is the main frequency component.

### 5.3. Bridge Vibration Monitoring Experiment

In this experiment, the target is the bridge cable. The photo of the experiment scene is shown in Figure 13. The radar we used is the same (TI) IWR1642 BoosterPack used in the vibrating calibrator experiment. The parameters are also the same.

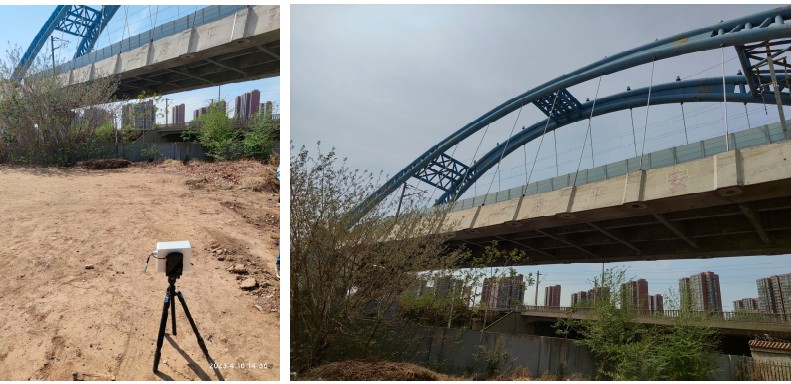

**Figure 13.** Photo of the bridge vibration monitoring experiment.

The experimental results are shown in Figure 14. Figure 14a shows the 1-D range profile radar signal. Multiple peaks in the figure correspond to cables with different ranges. In this experiment, the cable about 58.3 m away from the radar is selected as the experimental target. Figure 14b shows the Doppler signal with a base frequency of about 1.24 Hz, which indicates that the vibration frequency of the cable is about 1.24 Hz. Figure 14c shows the original vibration signal in the complex plane, and the estimated clutter is −0.16–0.08j. Figure 14d shows the deformation extracted from the original vibration signal, and the problem of jump error is serious. Figure 14e shows the vibration signal after periodic filtering. *M* in the periodic filter is set to two, so the SNR can theoretically be improved by five times. The estimated clutter is −0.10–0.03j. Figure 14f shows the deformation extracted from the filtered signal without jump error. The vibration amplitude is about 0.1 mm.

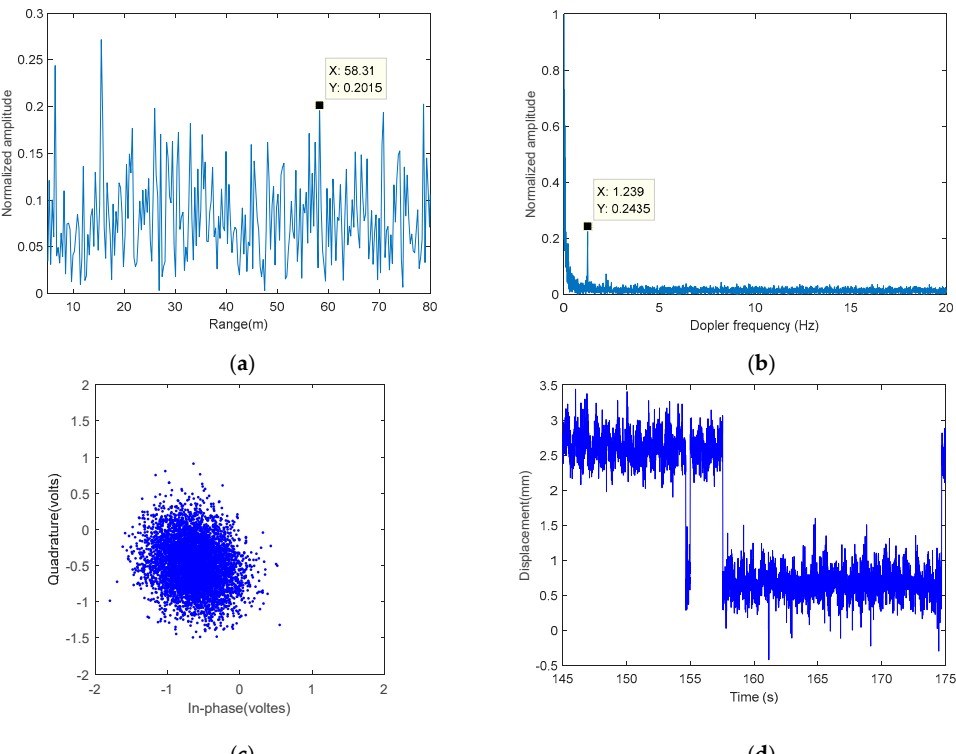

**Figure 14.** *Cont.*

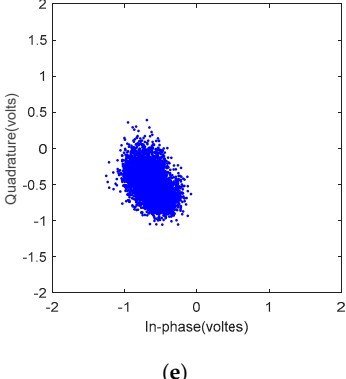
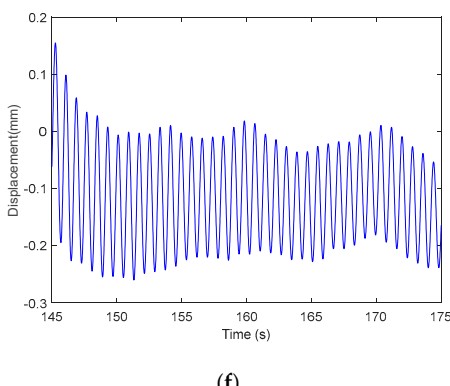

|         |         |
|:-------:|:-------:|
|   (e)   |   (f)   |

**Figure 14.** Bridge cable vibration monitoring results: (**a**) one-dimensional range profile; (**b**) Doppler signal; (**c**) original vibration signal in the complex plane; (**d**) deformation extracted from the original vibration signal; (**e**) complex plane vibration signal after periodic filtering; and (**f**) deformation extracted after periodic filtering.

## 6. Conclusions

In this paper, we propose a periodic filtering method for vibration radar signals and derive the expressions of the periodic filter in time and frequency domains. This method uses the periodic repetition characteristics of a vibration signal to accumulate several adjacent cycles and denoise the signal. Compared with the traditional low-pass filtering method, this method does not require oversampling for PRF. We verify the correctness and effectiveness of this method by simulation experiments, vibrating calibrator experiments, and bridge vibration monitoring experiments. The results show that when the PRF is not oversampled, the low-pass filtering method widely used in the existing literature will distort the vibration signal, so that the vibration deformation cannot be accurately retrieved. The proposed periodic filtering method, which does not require oversampling of the PRF, can significantly improve the SNR, thereby solving the problems of inaccurate clutter estimation and jump errors in deformation inversion. Therefore, parameters such as vibration frequency and amplitude can be accurately estimated.

This method has some limitations: it can only extract one frequency of vibration deformation from the signal at a time; for complex vibrations with multiple frequencies, each frequency has to be extracted separately. Our future work focuses on two aspects: (1) deriving a complex vibration filter for multiple-frequency vibrations and expanding the applicability of the method; and (2) studying the millimeter-wave multi-channel radar vibration measurement method. This system can image large infrastructures and acquire vibration frequencies and amplitudes at each scattering center. Combining imaging and vibration measurement facilitates vibration point localization. We aim to improve and optimize the proposed periodic filtering method and apply it to millimeter-wave multi-channel radar for high-precision vibration deformation extraction of large-scale infrastructure.

**Author Contributions:** Conceptualization, Y.L. (Yun Lin); methodology, Y.L. (Yun Lin) and L.Z.; software, Y.L. (Yun Lin) and L.Z.; validation, Y.L. (Yun Lin), L.Z. and H.H.; formal analysis, Y.L. (Yun Lin) and L.Z.; resources, W.S. and Y.L. (Yang Li); writing—original draft preparation, Y.L. (Yun Lin) and L.Z.; writing—review and editing, Y.L. (Yun Lin), L.Z., Y.L. (Yang Li), W.S. and Y.W.; supervision, Y.L. (Yun Lin); project administration, Y.W.; funding acquisition, Y.L. (Yun Lin) and Y.W. All authors have read and agreed to the published version of the manuscript.

**Funding:** This research was funded by the National Key R&D Program of China, grant number 2022YFF0606901; The National Natural Science Foundation of China, grant numbers 61860206013 and 62131001; and the Innovation Team Building Support Program of the Beijing Municipal Education Commission, grant number IDHT20190501.

**Data Availability Statement:** Not applicable.

**Acknowledgments:** We thank the anonymous reviewers for their good advice.

**Conflicts of Interest:** The authors declare no conflict of interest.

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
