# Peer review of "Periodic-Filtering Method for Low-SNR Vibration Radar Signal"

_remotesensing, doi:10.3390/rs15143461_

Round 1

Reviewer 1 Report (New Reviewer)

The manuscript can be major revision for many reasons but these can generally be divided into technical reasons.

1. Abstract should be precise and should clearly show the background, hypothesis, problem statements, methodology, techniques used with relevant method, and the results obtained.

2. Introduction section needs revision. It should also introduce some latest research results in the domain, and motivation for the proposed work. 

3. Literature review section must also be extended. Related works section may include literature survey.

4. The Limitations of the proposed study need to be discussed before conclusion.

5. What specific improvements should the authors consider regarding the methodology? What further controls should be considered?

6. Do you consider the topic original or relevant in the field? Does it address a specific gap in the field?

7. Are the conclusions consistent with the evidence and arguments presented and do they address the main question posed?

8. The contributions of this work need to be clearly articulated.

Language must be improved as there are linguistic errors at some places.

Author Response

Reviewer 2 Report (New Reviewer)

This paper has presented well. The contribution of this research is the periodic filtering method for denoising vibration radar signals and overcoming the problem of deformation jump error, so as to increase the accuracy of static clutter estimation. But there are some things that need to be explained further:

1. In the abstract it is necessary to present numerical results

2. It is necessary to discuss and present the increase in the accuracy of static clutter estimation after using the proposed method, it would be better if presented graphically as well

3. What are the advantages of the proposed method compared to other related work

4. It is necessary to explain in more detail the sampling dataset

Author Response

Reviewer 3 Report (New Reviewer)

The authors presented a periodic filtering method is proposed, which makes use of the periodic repetition characteristics of the vibration signal to filter and denoise the original vibration radar signal, and has no oversampling requirements for the PRF. After periodic filtering, static clutter can be estimated more accurately and the problem of deformation jump error can be solved.

The paper needs more improvements:

- The background in the abstract longer than the main work. You have to focus on your main work and contribution in the abstract.

- The related works are old, few refs only from recent years. So, add related work from recent three years, 2023, 2022, and 2021. Also, references format needs more modification. Check MDPI style.

- Also, comparison to similar methods can be considered ( but from most recent years).

- limitations and challenges can be further discussed, and accordingly, you can highlight some points for future directions.

- Proofreading is needed. 

Minor modifications can be made. 

Author Response

Reviewer 4 Report (New Reviewer)

Dear authors, thanks for the work proposed. It is very interesting from my point of view. 

Below you can find my comments about the paper:

1 - The first paragraph until citation 10 is not written clearly. I suggest to improve the quality of this part. 

2 - After citation 19, you assert that the phase jumps are random. I agree that this is a problem hard to solve but some papers in the literature say that this phenomenon is related to a changing of SNR. I suggest justifying this affirmation or remove from the paper.

3 - All the introduction is too general. I suggest better describing the state of the art, maybe with a particular focus on the radar used in the proposed work. 

4 - In section 2, the use of RVP must be better justified. 

5- In the same section, why did you perform the IFFT to extract the phase? Is not sufficient to take the argument of the sv?

6 - Section 3 is not supported by the experiments. If the clutter influences the quality of the measurements an experimental proof must be proposed. If this is already made in the literature please cite the reference of that work. 

7 - Flow chart: the proposed method assume that fv is known. What happens if the vibration signal is composed da many parts? how is possible to apply this approach when not only one vibration frequency is defined? If the method works even in this case please add also some experiments on that. 

8 -  Section 5 is not sufficiently described. Please improve the description of the radar system, the setup and the test proposed. 

9 - The calibrator used is already calibrated? if yes please provide a description of the calibration otherwise this must be made as without a reference is impossible to understand the accuracy of the proposed method. 

Round 2

Reviewer 2 Report (New Reviewer)

some revision points have been well done.

But in the abstract section I suggest that you need to be more assertive with the numeric numbers explained in the discussion/results, for example: an increase in SNR of up to five times, as in the discussion of the results

Author Response

Reviewer 3 Report (New Reviewer)

The authors addressed the comments. This version can be considered for publication. I have no more comments. 

Author Response

Reviewer 4 Report (New Reviewer)

Dear authors, thanks for the corrections made to the work. 

I think that the only critical point is the calibration. To demonstrate the capabilities of the system you need to use a calibrated vibrating target. 

In the conclusion part also a comparison between literature can be added. 

Author Response

This manuscript is a resubmission of an earlier submission. The following is a list of the peer review reports and author responses from that submission.

Round 1

Reviewer 1 Report

The paper proposed a periodic filtering method for vibration radar signal to solve the problems of inaccurate clutter estimation, and jump error in deformation inversion. In general, the structure of the paper is complete, which is innovative to some extent. It is recommended to review the manuscript after modification. The following problems still need to be explained:

1)     In line 326, what is the meaning of title1 and title2? Provide accurate definitions.

2)     In formula 18, how to determine the value of M in real conditions.

3)     There are some basic grammatical errors in the text such as in the 242-243 line.

4)     Simulation experiment and vibrating calibrator Experiment lack the result of using traditional low-pass filtering methods. Please add the relevant experiment. 

5)     Please add the comparison simulation experiment of your method and traditional low-pass filtering method under different SNR.

Reviewer 2 Report

Authors should modify or consider issues as follow:

  1. complex clutter environment , especially time variation clutter, clutter with micro or minor doppler,  should be considered, analysis, and processed in paper.
  2. The assumption that clutters in equ.10 are static is too ideal to believe.
  3. The filter in figure5(b) is obtained from a infinite time domain sequences which in real case  could not be obtained.
  4. From table 1, and assumption in section 5.1, it is obviously that Amplitude os vibration is greater than wavelength /4. Authors should indicate results when the assumption is failure and  compare results from different assumptions.
  5. The vertical axes should be given in dB scale in Figure 7 (b),(d),(h) and figure 9(b),(d),(h)
  6. To compare result with other instruments should be given for verifying the performance of mmw device mentioned in this paper.
  7. In section 5.3, the mmw device is deployed  on the same bridge to be monitored. So, the vibration of bridge will put its vibration on echoes of bridge obtained by the device , resulting experiment has less credibility. To redo experiment is suggested. 

Round 2

Reviewer 1 Report

The author has made serious replies and modifications according to the comments, and the manuscript is suggested for accepted.

Reviewer 2 Report

  1. Slowly changing clutter authors mentioned vary with continuous time , could not divided into time segment. 
  2. “Low SNR” does not define. From simulation, paper has shown SNR is 5dB in time domain with processing time 80s. However, For a stationary vibration signal within 80s, SNR with 5dB is quite strong condition to processing. So, authors should explain the limitation of “low SNR”. To give some results under 0dB SNR is strongly suggested.
  3. To convince reader, authors must compare result with other instruments which could be used as a bias device.
  4. Micro-doppler issue of target, clutter, platform and etc must be considered.